# Coenzyme A and Its Thioester Pools in Obese Zucker and Zucker Diabetic Fatty Rats

**DOI:** 10.3390/nu12020417

**Published:** 2020-02-06

**Authors:** Shigeru Chohnan, Shiori Matsuno, Kei Shimizu, Yuka Tokutake, Daisuke Kohari, Atsushi Toyoda

**Affiliations:** 1Department of Food and Life Sciences, Ibaraki University College of Agriculture, 3-21-1 Chuo, Ami, Ibaraki 300-0393, Japan; shiori.matsuno.afb@gmail.com (S.M.); sksk19900928@yahoo.co.jp (K.S.); daisuke.kohari.abw@vc.ibaraki.ac.jp (D.K.); atsushi.toyoda.0516@vc.ibaraki.ac.jp (A.T.); 2Department of Applied Life Science, United Graduate School of Agricultural Science, Tokyo University of Agriculture and Technology, 3-5-8 Saiwai, Fuchu, Tokyo 183-8509, Japan; y.tokutake.afb@gmail.com

**Keywords:** coenzyme A, acetyl-CoA, malonyl-CoA, Zucker rat, ZDF rat

## Abstract

Feeding behavior is closely related to hypothalamic malonyl-CoA level in the brain and diet-induced obesity affects total CoA pools in liver. Herein, we performed a comprehensive analysis of the CoA pools formed in thirteen tissues of Zucker and Zucker diabetic fatty (ZDF) rats. Hypothalamic malonyl-CoA levels in obese rats remained low and were almost the same as those of lean rats, despite obese rats having much higher content of leptin, insulin, and glucose in their sera. Regardless of the *fa*-genotypes, larger total CoA pools were formed in the livers of ZDF rats and the size of hepatic total CoA pools in Zucker rats showed almost one tenth of the size of ZDF rats. The decreased total CoA pool sizes in Zucker rats was observed in the brown adipose tissues, while ZDF-fatty rats possessed 6% of total CoA pool in the lean rats in response to *fa* deficiency. This substantially lower CoA content in the obese rats would be disadvantageous to non-shivering thermogenesis. Thus, comparing the intracellular CoA behaviors between Zucker and ZDF rats, as well as the lean and fatty rats of each strain would help to elucidate features of obesity and type 2 diabetes in combination with result (s) of differential gene expression analysis and/or comparative genomics.

## 1. Introduction

Coenzyme A (CoA) participates in numerous metabolic pathways as an acyl carrier in cells. This cofactor is synthesized through five enzymatic steps [1]. Pantothenate kinase (PanK) is a key enzyme catalyzing the first step in the production of 5’-phosphopantothenate. Four PanK homologs are encoded in mouse chromosomes and the expression pattern of the *PANK* genes is found to differ in tissues [1]. For example, *PANK1* and *PANK3* are prevalently expressed in murine liver, while *PANK2* and *PANK3* are predominant in the testis and brain [2]. Recently, human *PANK4* was revealed to be a pseudogene whose product does not display PanK activity [3]. Nevertheless, active PanKs provide CoA in tissues and are regulated by non-esterified CoA (CoASH) and acyl-CoAs. In addition, hypothalamic malonyl-CoA displayed a positive response to feeding behavior, leading to the depression of orexigenic neuropeptides and an increase in anorexigenic peptides [4]. As a result, this observation was applied to the feeding behavior in socially defeat-stressed mice [5]. Previously, we conducted a comprehensive analysis of CoA pools in thirteen tissues of Wistar rats [6,7]. The liver, heart, and brown adipose tissue have larger total CoA pools consisting of CoASH, acetyl-CoA, and malonyl-CoA than other tissues. In obese rats that are continuously fed a high-fat diet (HFD), the hepatic total CoA pool size shrank to one fifth of that of rats fed a high-carbohydrate diet (HCD) and a high-protein diet (HPD). However, CoA pools in the heart and brown adipose tissue were not affected by diet composition. In the brain, diet-induced obesity caused by HFD resulted in elevated malonyl-CoA level in the hypothalamus, indicating an energy-sufficient status. Furthermore, malonyl-CoA level exceeding that of acetyl-CoA was observed in the skeletal muscle of obese rats. Thus, CoA pools were found to be acutely influenced by the quality of diet and feeding, especially in pools of the hypothalamus, skeletal muscle, and liver.

Metabolic diseases, such as obesity and type 2 diabetes, are continuously increasing worldwide and have now become a serious problem. Model animal research contributes to the understanding of these diseases via basic and clinical studies. Because Zucker [8,9,10] and Zucker diabetic fatty (ZDF) [11] rats are widely used as model animals of obesity and type 2 diabetes, respectively, they were employed in the present study. Zucker rats have a missense mutation (*fa*) in the leptin receptor (*Lepr*) gene [12,13,14] and because of this point mutation, the homozygous (*fa*/*fa*) mutants exhibit hyperphagia, defective non-shivering thermogenesis, and energy deposition in the adipose tissue, which consequently lead to obesity [15]. In addition, obese rats display endocrinological defects such as insulin resistance, dyslipidemia, glucose intolerance, and hyperinsulinemia. ZDF rats are derived from the Zucker strain and exhibit non-insulin-dependent diabetes mellitus together with obesity [11,16]. Fatty rats are thus widely used in research on type 2 diabetes and they display insulin resistance, hyperglycemia, hyperlipidemia, moderate hypertension, and progressive renal injury [17].

In this study, we sought to provide information on acetyl-CoA, malonyl-CoA, CoASH, and the total CoA pools in 13 tissues of rats with genetically-induced obesity and diabetes mellitus. Additionally, we compared the data obtained from Zucker and ZDF rats.

## 2. Materials and Methods

### 2.1. Experimental Animals

Male Zucker-fatty (Crlj:Zuc-*Lepr^fa^*, *Lepr^fa^*/*Lepr^fa^*), Zucker-lean (Crlj:Zuc-*Lepr^fa^*, *Lepr^fa^*/+ or +/+), ZDF-fatty (ZDF-*Lepr^fa^*/CrlCrlj, *Lepr^fa^*/*Lepr^fa^*), and ZDF-lean (ZDF-*Lepr^fa^*/CrlCrlj, *Lepr^fa^*/+ or +/+) rats (6 rats/group; age, 11 weeks old) were purchased from Charles River Laboratories Japan Inc. (Kanagawa, Japan). Rats were individually housed in single cages (276 × 445 × 204 mm, polycarbonate, CLEA Japan, Inc., Tokyo, Japan) and were fed standard laboratory pellet chow (MF diet from Oriental Yeast Co. Ltd., Tokyo, Japan) and water ad libitum under a 12-h light (6:00 am to 6:00 pm)/12-h dark (6:00 pm to 6:00 am) cycle at 22 ± 1 °C. After a one-week acclimation period, rats were starved for 16 h with free access to drinking water. The cerebral cortex, hippocampus, hypothalamus, cerebellum, and medulla oblongata from the brain of rats and eight types of tissue, namely the liver, spleen, kidney (right side), heart, skeletal muscle (right soleus muscle), perirenal adipose tissue, brown adipose tissue, and epididymal adipose tissue, were removed. These tissues were stored at −80 °C until use.

This study was carried out in accordance with the guidelines of the Animal Care and Use Committee of Ibaraki University (Approval number 112) and the guidelines of the Ministry of Education, Culture, Sports, Science, and Technology, Japan (Notice No.71)

### 2.2. Extraction and Determination of the CoA Pool in Tissues

The prepared tissues were immersed in 400 µL of 0.6 M sulfuric acid per 100 mg of tissue, and homogenized [6,7]. Following storage of the extract at 4 °C overnight to inactivate the endogenous enzymes, the homogenized tissues were centrifuged at 9000× *g* at 4 °C for 10 min, and 1 M Tris (1/20 volume of supernatant) was added to the supernatant. The acidic extract was adjusted to approximately pH 6.5 with NaOH on ice. Thereafter, the solution was maintained at −80 °C overnight. After thawing, precipitation was removed by centrifugation, and acetyl-CoA, malonyl-CoA, and CoASH in the extract were determined using the acyl-CoA cycling method described below [18,19].

All four enzymes used in this method were prepared in our laboratory. Malonate decarboxylase, a multifunctional enzyme, was partially purified from *Pseudomonas putida* JCM 20089 (formerly IAM 1177) [20,21]. The other enzymes, namely acetate kinase (AckA from *Escherichia coli* W3110), citrate synthase (CitZ from *Bacillus subtilis* str. 168), and phosphate acetyltransferase (Pta from *B. subtilis* str. 168), were cloned and expressed in *E. coli* JM109 as His-tagged enzymes [22]. The reaction mixture used to measure the sum of acetyl-CoA and malonyl-CoA contained 50 mM Tris-HCl (pH 7.2), 1 mM 2-mercaptoethanol, 10 mM MgSO_4_, 50 mM malonate, 10 mM ATP, 1 U of malonate decarboxylase, and the tissue extracts containing acetyl-CoA and/or malonyl-CoA (2.5 to 80 pmol) in 400 µL. The cycling reaction was initiated by adding malonate decarboxylase. After a 20-min incubation at 30 °C, 1 U of the recombinant acetate kinase was added to the mixture. After an additional 20 min, 0.2 mL of 2.5 M neutralized hydroxylamine was added, and the incubation was continued for an additional 20 min at 30 °C. The reaction was terminated by adding 0.6 mL of 10 mM ferric chloride dissolved in 25 mM trichloroacetic acid-1 M HCl. The *A*_540_ of the acetohydroxamate formed was measured. Each assay was performed in duplicate.

Malonyl-CoA and CoASH in the tissue extracts were separately determined by eliminating acetyl-CoA with citrate synthase (CS) and converting CoASH to acetyl-CoA with phosphate acetyltransferase (PAT), respectively, prior to carrying out measurements with the acyl-CoA cycling method. The reaction mixture for CS treatment contained 50 mM Tris-HCl (pH 7.2), 2 mM oxaloacetate, 5 µg of the recombinant CS, and the tissue extracts in 1 mL. The reaction was carried out at 25 °C for 20 min and terminated by placing the reaction tube on an ice slush. One milliliter reaction mixture for PAT treatment contained 50 mM Tris-HCl (pH 7.2), 0.1 mM acetylphosphate, 0.8 µg of the recombinant His-tagged PAT, and the tissue extracts. After incubation at 25 °C for 20 min, the reaction was terminated by eliminating the His-tagged enzyme through Ni sepharose 6 Fast Flow (GE Healthcare UK Ltd., Buckinghamshire, UK). The remaining malonyl-CoA in the CS treatment and the acetyl-CoA derived from CoASH were determined using the acyl-CoA cycling method.

### 2.3. Analysis of Blood Serum

Glucose, triglycerides, non-esterified fatty acids (NEFAs), total ketone bodies (T-KB), total cholesterol (T-CHO), insulin, and leptin in the sera of Zucker and ZDF rats were analyzed at Nagahama Life Science Laboratory (Oriental Yeast Co. Ltd., Shiga, Japan). Briefly, glucose, triglycerides, NEFAs, T-KB, and T-CHO were assayed by enzymatic methods while insulin and leptin were measured by ELISA.

### 2.4. Statistical Analysis

Body weight, tissue weight, blood serum, and CoA pools in each tissue were compared between Zucker and ZDF, lean and fatty rats, and their interaction by Two-way ANOVA using R Ver. 3.5.2 (R Development Core Team). Statistical significance was defined as *p* < 0.05.

## 3. Results

### 3.1. Analysis of Body Weight, Tissue Weight, and Blood Serum

As shown in Table 1, Zucker- and ZDF-fatty rats exhibited hyperphagia. Compared to lean rats, daily food intake and drinking by Zucker-fatty rats were 1.7- and 1.8-fold higher during one week of habitation, thereby leading to obesity. In ZDF-fatty rats, the level of food intake per day was 2.4-fold higher than that in lean rats. Additionally, daily drinking was markedly elevated to a value 5.0-fold higher (151 ± 4 mL/day). Body weight of Zucker-fatty rats was clearly increased, while ZDF-fatty rats had a weight loss of 6.72 ± 1.37 g during one week of habitation.

The livers, kidneys, perirenal adipose tissues, brown adipose tissues, and epididymal adipose tissues of both fatty rats were enlarged; however, no differences were found between fatty and lean rats in the spleen and heart. The increased weights observed in the three adipose tissues from fatty rats were more marked in Zucker-fatty rats while the gain in kidney was more prominent in ZDF-fatty rats. The effect of *fa* deficiency on skeletal muscles differed between the two strains and the weight of those from Zucker-fatty rats was unambiguously decreased.

Blood serum analysis of Zucker and ZDF rats revealed that both fatty rats had high levels of leptin (Table 2). In particular, Zucker-fatty rats had a leptin level of 73.0 ± 3.4 ng/mL, which was 35-fold higher than that in lean rats. Despite starving, the blood glucose level in Zucker-fatty rats was substantially higher (251 ± 9 mg/dL) than that in lean rats. As a result, their insulin level was 7.9-fold higher than that of lean rats. In contrast, ZDF-fatty rats had an insulin level of 2.06 ± 0.75 ng/mL, which was as low as that of Zucker-lean rats. Conversely, their blood glucose level was as high as that of Zucker-fatty rats. Serum lipids such as triglycerides, NEFAs, and T-CHO were much higher in obese rats than lean rats. Additionally, a significant difference was found between fatty and lean rats in T-KB levels. Thus, Zucker-fatty rats presented typical obesity with hyperglycemia, hyperlipidemia, hyperinsulinemia, and hyperleptinemia while ZDF-fatty rats exhibited symptoms of abnormal polydipsia caused by diabetes as shown in Table 1. The values of these metabolic parameters in the bred Zucker and ZDF rats were close to the results previously reported by Jonas et al. [23].

### 3.2. Analysis of CoA Pools

The sizes of acetyl-CoA, malonyl-CoA, and CoASH, and the total CoA pools (sum of acetyl-CoA, malonyl-CoA, and CoASH) in the brain tissues of Zucker and ZDF rats were analyzed (Table 3). Although each CoA pool in the brain tissues of the Zucker and ZDF obese rats exhibited almost the same or a lower size than the respective lean rats, the medulla oblongata in the Zucker-fatty rats and the cerebral cortex in the ZDF-fatty rats clearly had larger total CoA pools. Conversely, the pool sizes of the cerebral cortex in the Zucker-fatty rats and the medulla oblongata in the ZDF-fatty rats were smaller. By comparing the total CoA pools between both strains, ZDF rats were found to possess a larger total CoA pool (i.e., approximately 10 nmol/mg or higher) in tissues, except in the hypothalamus. In the hypothalamus, the acetyl-CoA pools were larger in ZDF rats but markedly decreased in obese rats. Conversely, the malonyl-CoA pools, which are closely related to feeding behavior, had almost the same values in lean and fatty rats, although obese rats exhibited hyperphagia.

The liver, heart, and brown adipose tissue of Wistar rats had substantially larger total CoA pools (i.e., exceeding 10 nmol/g of tissue) among the thirteen tissues measured [6,7]. In particular, the liver could form a large total CoA pool of approximately 100 nmol/g of tissue under starvation in HCD-fed rats [7]. The present study revealed that the total CoA pools were substantially reduced in Zucker rats. In contrast, ZDF rats had large total CoA pools of 88.2 ± 10.5 nmol/g and 114 ± 8 nmol/g in the tissue of lean and fatty rats, respectively, owing to the remarkably high levels of CoASH. These values were close to those of the livers of the HCD- or HPD-fed Wistar rats. In addition to the clearly enlarged liver recognized in obese rats, their total CoA content was found to be higher than that in lean rats. The phenomena of no significant difference in total CoA pools between lean rats and fatty rats and the considerably larger sizes in ZDF than Zucker rats were also observed in the heart. The characteristic of *fa*-deficiency was found in the total CoA pool for the brown adipose tissues of ZDF rats. The content of acetyl-CoA, malonyl-CoA, and CoASH in fatty rats were much lower, with 26.8%, 36.2%, and 4.5% of the values found in lean rats, respectively. Consequently, the total CoA pool was reduced to 6.0%. These low content of acylated CoAs were close to the respective pool sizes in the epididymal adipose tissue, which is one of the tissues that had the lowest CoA contents together with the perirenal adipose tissue. Thus, it is noteworthy that all CoA pools in the brown adipose tissue of ZDF-fatty rats were abnormally reduced. Although the reduction of the total CoA pool was observed in Zucker obese rats, originally, the total CoA pool in the lean rats had a small size of 6.01 ± 0.69 nmol/g of tissue.

In skeletal muscles, the malonyl-CoA pools were larger than the acetyl-CoA pools in all four rats. The higher contents of malonyl-CoA might interfere with the incorporation of fatty acids by carnitine acyltransferase I into the mitochondria and the subsequent β-oxidation. Because free CoA was not detected in the spleens of Zucker rats and ZDF-fatty rats, reduced sizes of the total CoA pools were observed. In the kidneys, the total CoA pools were not affected by *fa*-deficiency and strain.

## 4. Discussion

In the hypothalamus, malonyl-CoA functions as a mediator that controls feeding behavior by downregulating orexigenic neuropeptides (neuropeptide Y and agouti-related peptide) and upregulating anorexigenic neuropeptides (proopiomelanocortin and cocaine- and amphetamine regulated-transcript) [4]. Malonyl-CoA is modulated by the phosphorylation and dephosphorylation of acetyl-CoA carboxylase (ACC), which is one of the target proteins of AMP-activated protein kinase (AMPK) [24]. Leptin is known to activate ACC by inhibiting AMPK activity and then enhance hypothalamic malonyl-CoA level in mice [25,26]. Thus, leptin, which is secreted from white adipose tissues, suppresses food intake through the hypothalamic malonyl-CoA signaling system described above. In the present study, Zucker- and ZDF-fatty rats, which have a deficient leptin receptor, have a largely excess amount of leptin in their blood sera (Table 2). However, almost the same sizes of malonyl-CoA pools were observed in the *fa*/*fa* Zucker and ZDF strains as those of the respective lean rats (Table 3). The results obtained with the leptin receptor-deficient rats supported the finding that the governance of malonyl-CoA content by phosphorylation and dephosphorylation of AMPK in the hypothalamus occurs mainly downstream of the leptin signal. AMPK is rigorously regulated by the intracellular level of AMP, which is increased in the energy-starved state [27]. Wolfgang and his co-workers reported that the regulation of AMPK and ACC via the response to glucose in *ob*/*ob* mice still functioned in a leptin-independent manner [26]. Therefore, the hypothalamic malonyl-CoA levels in these fatty rats were as low as those in lean rats, suggesting that although brain cells have higher levels of blood glucose, cells in the hypothalamus of obese rats could detect energy starvation. This phenomenon would indicate the inhibition of glucose incorporation via insulin-resistance in the hypothalamus. Insulin receptors are expressed in the hypothalamus [28], while glucose transporter 2 (GLUT2) mainly mediates glucose incorporation with the insulin-independent system [29,30]. Unlike *ob*/*ob* mice, hypothalamic malonyl-CoA behavior alone cannot explain the relationship between energy balance and appetite in *fa*-deficient rats that display insulin resistance in addition to leptin resistance. Neuropeptide Y is known to be abundant in obese Zucker rats [31,32,33]. Hence, the regulation of hypothalamic malonyl-CoA level through AMPK, which recognizes the change in energy status, may be disrupted, and the increase in ghrelin may predominantly function as a mediator of hyperphagia in Zucker rats [34,35].

The hepatic total CoA pool, mainly the CoASH constituent, is increased owing to its use in *β*-oxidation in the fasted state [6,36]. The livers of Wistar rats fed HFD for 4 weeks were enlarged while their hepatic total CoA pools were greatly reduced to almost one fifth of those of the HCD-fed rats according to the decrease in the CoASH pool [7]. These findings imply that hepatic CoA metabolism was greatly affected by the quality of the diets and body energy states. In the present study, hypertrophy was observed in the livers of Zucker- and ZDF-fatty rats (i.e., the genetically-induced obese rats) (Table 1). Interestingly, an extremely small size of the total CoA pools (i.e., approximately 10 nmol/g of tissue) was found in Zucker-lean and fatty rats (Table 3). This observation might be an intrinsic characteristic of hepatic metabolism in Zucker rats, regardless of the *fa* genotype. The hepatic total CoA pool sizes in both ZDF-obese and lean rats had nearly 10-fold higher values than Zucker rats (Table 3). Thus, it is interesting that Zucker rats had hepatic total CoA pools similar to Wistar obese rats fed an HFD. ZDF rats also had almost the same size as Wistar rats fed HCD, regardless of their fatty or lean appearance. In addition, the compositions of the CoASH and acyl-CoA contents in Zucker and ZDF rats resembled those of Wistar rats fed HFD and HCD, respectively [7]. Based on the CoA metabolites, the hepatic metabolism of Zucker and ZDF rats displayed inherent differences. At the molecular level, the hepatic total CoA pool size is balanced with CoA synthesis by PanK1, the mainly expressed PanK isoform in murine liver [37,38], and its degradation by nudix hydrolase 7 [39,40,41]. These enzyme activities are modulated by the action of peroxisome proliferator-activated receptor α (PPARα) [41,42,43]. Hepatic PPARα content in Zucker obese rats is statistically lower than that of lean rats [44]. However, Zucker obese rats and lean rats displayed an equivalent total CoA pool. Although the total CoA pool size was much larger in ZDF rats than Zucker rats, a similar result was obtained between ZDF and Zucker rats. Analysis of the hepatic CoA pools in Zucker and ZDF rats revealed a discrepancy between in vivo CoA behavior and the response of PPARα to CoA biosynthesis on fatty acid degradation in the fasted state. On the other hand, hepatic hypertrophy and hyperlipidemia were observed in both obese rats, and concomitantly, serum leptin and insulin were higher in fatty rats than in lean rats (Table 1 and Table 2). It has been reported that leptin inhibits the production of triacylglycerol [45], whereas insulin promotes it [46,47]. In obese Zucker rats (*fa*/*fa*), approximately 2 to 4 times more triacylglycerol than in lean rats (+/+ or *fa*/+) was accumulated in the liver, followed by its secretion into circulation [48]. Since the leptin receptor did not function in the fatty rats, the turnover of triacylglycerol was more active in the synthesis than in the degradation by the action of insulin. Consequently, the accumulation of triacylglycerol was observed in the liver of obese rats. In the present study, this phenomenon occurred in ZDF-fatty rats as well as in Zucker-fatty rats, and this result reflected the amounts of insulin in sera well. In addition, the significantly lower total CoA levels in Zucker rats were clearly disadvantageous for β-oxidation, indicating that Zucker rats were in an in vivo environment that favored the accumulation of triacylglycerol (Table 3). Previous findings did not reveal the hepatic CoA metabolism and hypothalamic CoA metabolism in these obese rats.

Zucker- and ZDF-fatty rats demonstrated progressive renal failure [49,50,51,52]. Although renal hypertrophy was observed in these obese rats, there was no important difference between lean and fatty rats in CoA contents, suggesting that renal intracellular metabolisms were seldom damaged, at least until 12-weeks old (Table 3).

Malonyl-CoA is often present as a minor CoA molecular species in cells. However, its level was found to exceed that of acetyl-CoA in the skeletal muscles of HFD-fed obese Wistar rats [7]. Herein, we found the same result in the skeletal muscles of Zucker and ZDF rats. This predominance of malonyl-CoA relative to acetyl-CoA in skeletal muscles might be established by other genetic factor (s), such as deficiency of liver kinase B1 (LKB1) or AMPK, and not the obesity induced by their *fa*-deficiencies. This is because even lean rats possessed larger malonyl-CoA than acetyl-CoA pools. LKB1 is a major AMPK kinase in skeletal muscle [53,54]. Accordingly, phosphorylated-AMPK levels were reduced and ACCs were activated in LKB1 knockout and dominant negative mice [54,55,56].

Compared to HCD-fed rats, diet-induced obesity in Wistar rat resulted in unchanged total CoA pool size in the brown adipose tissue [7]. Here, we found that genetically-induced obesity affected the total CoA pool sizes in brown adipose tissues (Table 3). The Zucker and ZDF obese rats had extremely small pools of total CoA (4.10 ± 0.87 and 2.82 ± 0.48 nmol/g of tissue) and acetyl-CoA (0.406 ± 0.102 and 0.394 ± 0.044 nmol/g of tissue), while Wistar rats tended to form large pools of total CoA (20–60 nmol/g of tissue) and acetyl-CoA (5–10 nmol/g) [6,7]. Zucker-lean rats possessed a small total CoA pool similar to Zucker-fatty rats, implying that those with the *fa*/*+* or *+*/*+* had already acquired abnormal CoA metabolisms in their brown adipose tissue and liver, which were subsequently manifested by the leptin-receptor deficient rats. PPARα was found to be abundant in the brown adipose tissue of Zucker rats. Conversely, obese rats expressed the receptor at a significantly lower level than lean rats [44]. This finding is inconsistent with our observation in Zucker rats, but aligned well with the relationship between ZDF-lean and fatty rats based on CoA behaviors. Brown adipose tissue plays a crucial role in thermogenesis via the mitochondrial proton conductance pathway mediated by uncoupling protein 1. The enlarged brown adipose tissue of Zucker-fatty rat displayed reduced activity in the mitochondrial proton conductance pathway [57]. Our data demonstrate that the substantially smaller acetyl-CoA pool in Zucker-fatty rat provided a further disadvantage to this non-shivering thermogenesis. This is because NADH and FADH_2_ could form an imbalance in H^+^ between the intermembrane space and matrix in mitochondria via electron transport, which is generated from acetyl-CoA through the TCA cycle.

PanK was recently reported to be closely related to cellular CoA biosynthesis, as well as indirectly related to insulin sensitivity and glucose homeostasis via micro RNAs 103 and 107 located in intron 5 of the three *PANK* genes [58,59,60]. Therefore, CoASH, acetyl-CoA, and malonyl-CoA are key metabolic regulators in addition to their role as intermediates in many pathways.

## 5. Conclusions

Zucker- and ZDF-fatty rats are widely used as the respective animal models of obesity and diabetes caused by mutation of the leptin receptor. Although ZDF rats are derived from Zucker rats, the sizes of their hepatic total CoA pool and the contents of their three molecular species were found to considerably differ. In addition, a negative influence on CoA behavior was found in the brown adipose tissue of *fa*/*fa* rats alone. Conversely, Zucker rats, even those that were lean and displayed the *fa*/+ or +/+ genotype, displayed abnormal CoA metabolism in their livers and brown adipocytes. Based on CoA contents, the obesity in Zucker rats cannot be clearly explained by functional deficiency in the leptin receptor alone. In the future, not only the comparison between Zucker-fatty and Zucker-lean rats, and ZDF-fatty and ZDF-lean rats (i.e., intra-strain comparison), but also that between Zucker-lean and ZDF-lean rats, and Zucker-fatty and ZDF-fatty rats (inter-strain comparison) on genetical and physiological data will be important for elucidating obesity and type 2 diabetes mellitus. Moreover, monitoring of the in vivo behaviors of CoASH and its derivatives in the hypothalamus, liver, skeletal muscle, and brown adipose tissue together with differential gene expression analysis and/or comparative genomics will aid in the rational explanation of obesity and type 2 diabetes in the respective Zucker and ZDF rat models.

## Figures and Tables

**Table 1 nutrients-12-00417-t001:** Body weight, food intake, drinking, and tissue weights in rats.

	Zucker		ZDF		Two-way ANOVA		
	Lean	Fatty	Lean	Fatty	Str.	*fa*	Str. × *fa*
Body weight (g)							
11 weeks	305.3 ± 5.3	429.0 ± 2.9	286.2 ± 2.2	344.7 ± 5.8	***	***	***
12 weeks	327.5 ± 6.8	459.6 ± 2.7	297.2 ± 2.1	338.0 ± 5.6	***	***	***
Body weight gain (g/week)	22.2 ± 2.0	30.5 ± 1.9	10.9 ± 0.8	−6.72 ± 1.37	***	**	***
Food intake (g/day)	23.3 ± 1.0	39.1 ± 1.0	17.0 ± 0.2	41.5 ± 2.8	*n*s	***	*
Drinking (mL/day)	29.6 ± 1.7	52.0 ± 5.9	30.1 ± 0.7	151 ± 4	***	***	***
Liver (g)	9.53 ± 0.46	18.1 ± 1.0	8.50 ± 0.08	16.5 ± 0.5	*	***	*n*s
Spleen (g)	0.459 ± 0.020	0.490 ± 0.029	0.524 ± 0.012	0.504 ± 0.017	*n*s	*n*s	*n*s
Kidney (g)	1.23 ± 0.03	1.41 ± 0.04	1.18 ± 0.05	1.65 ± 0.05	*	***	**
Heart (g)	0.999 ± 0.056	1.06 ± 0.03	1.13 ± 0.04	1.11 ± 0.03	*	*n*s	*n*s
Skeletal muscle (g)	0.171 ± 0.008	0.128 ± 0.007	0.144 ± 0.003	0.136 ± 0.004	*n*s	***	**
Perirenal adipose tissue (g)	3.79 ± 0.23	18.4 ± 0.7	1.64 ± 0.09	9.35 ± 0.33	***	***	***
Brown adipose tissue (g)	0.533 ± 0.031	1.76 ± 0.09	0.344 ± 0.027	0.782 ± 0.048	***	***	***
Epididymal adipose tissue (g)	4.65 ± 0.31	14.2 ± 0.3	2.58 ± 0.08	6.33 ± 0.36	***	***	***

All data are expressed as mean ± SEM (*n* = 6). ZDF - Zucker diabetic fatty. Statistical significances between Zucker and ZDF (Str.), lean and fatty rats (*fa*), and their interaction (Str. × *fa*) were derived by two-way ANOVA: *n*s, not significant; *, *p* < 0.05; **, *p* < 0.01; ***, *p* < 0.001.

**Table 2 nutrients-12-00417-t002:** Biochemical analysis of blood serum.

	Zucker		ZDF		Two-way ANOVA		
	Lean	Fatty	Lean	Fatty	Str.	*fa*	Str. × *fa*
Glucose (mg/dL)	140 ± 4	251 ± 9	114 ± 5	295 ± 20	ns	***	**
Triglycerides (mg/dL)	40.7 ± 3.5	542 ± 59	19.8 ± 1.5	237 ± 40	***	***	***
NEFAs (μEq/L)	286 ± 29	604 ± 61	261 ± 14	598 ± 39	ns	***	ns
T-KB (μmol/L)	830 ± 36	1032 ± 233	726 ± 45	1170 ± 186	ns	*	ns
T-CHO (mg/dL)	67.7 ± 4.1	102 ± 5	64.3 ± 1.0	137 ± 6	**	***	**
Insulin (ng/mL)	1.54 ± 0.19	12.1 ± 3.9	0.602 ± 0.072	2.06 ± 0.75	*	**	*
Leptin (ng/mL)	2.11 ± 0.07	73.0 ± 3.4	0.595 ± 0.034	12.7 ± 1.7	***	***	***

All data are expressed as mean ± SEM (*n* = 6). Statistical significances between Zucker and ZDF (Str.), lean and fatty rats (*fa*), and their interaction (Str. × *fa*) were derived by two-way ANOVA: ns, not significant; *, *p* < 0.05; **, *p* < 0.01, ***, *p* < 0.001. NEFAs, non-esterified fatty acids; T-KB, total ketone bodies; T-CHO, total cholesterol.

**Table 3 nutrients-12-00417-t003:** CoA pools in rat tissues.

	CoA	Zucker		ZDF		Two-way ANOVA		
Tissues	Species	Lean	Fatty	Lean	Fatty	Str.	*fa*	Str. × *fa*
Cerebral	A	2.16 ± 0.11	1.55 ± 0.06	2.55 ± 0.21	2.07 ± 0.12	**	**	ns
cortex	M	0.267 ± 0.018	0.230 ± 0.006	0.299 ± 0.035	0.155 ± 0.011	ns	***	*
	CoA	6.09 ± 1.04	4.52 ± 1.22	5.62 ± 1.47	13.2 ± 0.5	**	*	***
	Total	8.52 ± 1.11	6.29 ± 1.19	8.48 ± 1.66	15.4 ± 0.6	**	ns	**
Hippocampus	A	1.64 ± 0.17	1.39 ± 0.09	2.34 ± 0.08	2.14 ± 0.09	***	ns	ns
	M	0.243 ± 0.014	0.214 ± 0.015	0.276 ± 0.021	0.211 ± 0.014	ns	**	ns
	CoA	1.04 ± 0.22	1.47 ± 0.19	11.7 ± 0.6	9.58 ± 1.90	***	ns	ns
	Total	2.92 ± 0.37	3.08 ± 0.27	14.3 ± 0.7	11.9 ± 2.0	***	ns	ns
Hypothalamus	A	1.36 ± 0.10	1.14 ± 0.07	2.29 ± 0.07	1.49 ± 0.08	***	***	**
	M	0.173 ± 0.015	0.152 ± 0.012	0.137 ± 0.006	0.130 ± 0.007	*	ns	ns
	CoA	1.31 ± 0.18	1.06 ± 0.10	1.86 ± 0.20	1.06 ± 0.16	ns	**	ns
	Total	2.84 ± 0.27	2.35 ± 0.15	4.29 ± 0.23	2.68 ± 0.22	***	***	*
Cerebellum	A	3.74 ± 0.19	2.83 ± 0.07	2.68 ± 0.16	2.25 ± 0.17	*	**	ns
	M	0.511 ± 0.042	0.480 ± 0.025	1.54 ± 0.06	1.52 ± 0.06	***	ns	ns
	CoA	2.55 ± 0.45	1.31 ± 0.11	10.9 ± 1.0	9.33 ± 0.94	***	ns	ns
	Total	6.80 ± 0.65	4.62 ± 0.12	15.1 ± 1.0	13.1 ± 1.0	***	**	ns
Medulla	A	1.02 ± 0.25	1.31 ± 0.12	2.44 ± 0.04	1.76 ± 0.05	***	ns	**
oblongata	M	0.074 ± 0.017	0.119 ± 0.020	0.264 ± 0.013	0.187 ± 0.014	***	ns	**
	CoA	2.32 ± 1.24	7.83 ± 1.86	6.61 ± 1.48	4.67 ± 1.13	ns	ns	*
	Total	3.40 ± 1.46	9.26 ± 1.93	9.31 ± 1.48	6.62 ± 1.14	ns	ns	*
Liver	A	3.28 ± 0.39	2.37 ± 0.15	5.78 ± 0.98	5.83 ± 0.37	***	ns	ns
	M	0.918 ± 0.103	0.745 ± 0.077	1.58 ± 0.20	2.42 ± 0.17	***	*	**
	CoA	6.15 ± 2.32	9.20 ± 2.72	80.8 ± 9.4	105 ± 8	***	*	ns
	Total	10.4 ± 2.6	12.3 ± 2.8	88.2 ± 10.5	114 ± 8	***	ns	ns
Spleen	A	0.740 ± 0.066	0.686 ± 0.157	1.27 ± 0.21	0.993 ± 0.114	*	ns	ns
	M	1.37 ± 0.12	1.17 ± 0.15	1.37 ± 0.10	1.14 ± 0.04	ns	ns	ns
	CoA	nd	nd	2.36 ± 0.93	nd	–	–	–
	Total	2.11 ± 0.15	1.86 ± 0.26	5.00 ± 1.09	2.14 ± 0.11	*	*	*
Kidney	A	1.09 ± 0.06	0.791 ± 0.090	1.90 ± 0.10	1.82 ± 0.11	***	ns	ns
	M	0.561 ± 0.048	0.567 ± 0.026	0.865 ± 0.038	0.909 ± 0.046	***	ns	ns
	CoA	2.12 ± 0.38	1.65 ± 0.55	1.53 ± 0.28	1.18 ± 0.31	ns	ns	ns
	Total	3.78 ± 0.44	3.00 ± 0.60	4.30 ± 0.30	3.91 ± 0.43	ns	ns	ns
Heart	A	4.79 ± 0.56	3.21 ± 0.64	11.7 ± 0.6	8.93 ± 0.35	***	**	ns
	M	2.66 ± 0.45	1.47 ± 0.34	0.703 ± 0.028	0.699 ± 0.043	***	*	*
	CoA	15.9 ± 3.5	24.7 ± 5.3	42.1 ± 4.5	31.9 ± 2.8	***	ns	*
	Total	23.3 ± 4.1	29.4 ± 6.0	54.4 ± 5.0	41.5 ± 3.1	***	ns	ns
Skeletal	A	0.993 ± 0.127	0.660 ± 0.117	0.703 ± 0.113	1.09 ± 0.14	ns	ns	**
muscle	M	1.52 ± 0.16	1.61 ± 0.13	1.94 ± 0.28	2.00 ± 0.13	*	ns	ns
	CoA	5.89 ± 0.43	6.26 ± 0.80	6.28 ± 1.08	9.70 ± 0.99	*	*	ns
	Total	8.40 ± 0.43	8.53 ± 0.82	8.93 ± 1.13	12.8 ± 1.0	*	*	*
Perirenal	A	0.103 ± 0.025	0.042 ± 0.011	0.070 ± 0.018	0.016 ± 0.003	ns	**	ns
adipose tissue	M	0.191 ± 0.050	0.166 ± 0.030	0.197 ± 0.023	0.069 ± 0.016	ns	*	ns
	CoA	0.395 ± 0.094	0.211 ± 0.044	0.398 ± 0.150	0.219 ± 0.108	ns	ns	ns
	Total	0.690 ± 0.152	0.418 ± 0.077	0.666 ± 0.174	0.304 ± 0.125	ns	*	ns
Brown	A	1.44 ± 0.29	0.406 ± 0.102	1.47 ± 0.15	0.394 ± 0.044	ns	***	ns
adipose tissue	M	0.916 ± 0.212	0.260 ± 0.079	1.14 ± 0.06	0.413 ± 0.023	ns	***	ns
	CoA	3.65 ± 0.66	3.43 ± 1.01	44.4 ± 5.3	2.02 ± 0.43	***	***	***
	Total	6.01 ± 0.69	4.10 ± 0.87	47.0 ± 5.4	2.82 ± 0.48	***	***	***
Epididymal	A	0.186 ± 0.074	0.021 ± 0.005	0.233 ± 0.035	0.208 ± 0.027	*	*	ns
adipose tissue	M	0.114 ± 0.052	0.020 ± 0.009	0.200 ± 0.021	0.184 ± 0.010	***	ns	ns
	CoA	0.253 ± 0.153	0.041 ± 0.015	0.475 ± 0.082	0.435 ± 0.100	**	ns	ns
	Total	0.553 ± 0.267	0.082 ± 0.021	0.908 ± 0.133	0.827 ± 0.109	**	ns	ns

“A”, “M”, “CoA”, and “Total” indicate acetyl-CoA, malonyl-CoA, CoASH, and total CoA (defined as the sum of the three CoA pools), respectively. All data are expressed as nmol/g of tissue and mean ± SEM (*n* = 6). Statistical significance between Zucker and ZDF (Str.), lean and fatty rats (*fa*), and their interaction (Str. **×**
*fa*) were derived by two-way ANOVA: ns, not significant; *, *p* < 0.05; **, *p* < 0.01; ***, *p* < 0.001. nd, not detected.

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
