# Peer review of "Coenzyme A and Its Thioester Pools in Obese Zucker and Zucker Diabetic Fatty Rats"

_nutrients, 2020, doi:10.3390/nu12020417_

Round 1

Reviewer 1 Report

REVIEWER COMMENTS ON nutrients-706351

Comments to the authors:

Did the authors quantified the liver total fat or free fatty acids content in the liver samples? And also, did they measured the different lipoprotein-transported cholesterol from serum? If so they should include those parameters in the results and discussion. If not, it would be interesting to approach that information based on literature about how the liver fat, and liver fat composition affects the total hepatic CoA pool and the contents of their three molecular species in liver tissue, and then how the lipoprotein metabolism changes could affect the CoA levels and distribution among tissues, given the fact that this animal models are closely related to atherogenic processes and lipids variation could directly affect the CoA expressions.

Author Response

#nutrients-706351

January 29, 2020

Dear Dr. Reviewers:

   Thank you very much for your Review Reports of January 20, 2020 concerning our manuscript (nutrients-706351: Coenzyme A and Its Thioester Pools in Obese Zucker and Zucker Diabetic Fatty Rats). I greatly appreciate all of your suggestions for improvement to this manuscript.  We closely examined these comments and we have made the necessary corrections. The revised manuscript, included as a separate attachment, is a great improvement over the original.

                                                                                                        Sincerely yours,

Shigeru Chohnan, Ph.D.

 Ibaraki University College of Agriculture,

3-21-1 Chuo, Ami, Ibaraki 300-0393, JAPAN

 Tel & Fax: +81-29-888-8672

Our incorporation of the reviewers' suggestions is as follows:

Reviewer 1-Comment to Author:

Comment: Did the authors quantified the liver total fat or free fatty acids content in the liver samples? And also, did they measured the different lipoprotein-transported cholesterol from serum? If so they should include those parameters in the results and discussion. If not, it would be interesting to approach that information based on literature about how the liver fat, and liver fat composition affects the total hepatic CoA pool and the contents of their three molecular species in liver tissue, and then how the lipoprotein metabolism changes could affect the CoA levels and distribution among tissues, given the fact that this animal models are closely related to atherogenic processes and lipids variation could directly affect the CoA expressions.

Response: We did not estimate total fat or free fatty acids content in the liver and did not conduct the measurement of lipoprotein-transported cholesterol from serum, either. Therefore, we added the below sentences in Discussion section, line 263-275, with reference to papers reporting the lipid content in the liver of rats (references 45-48).

On the other hand, hepatic hypertrophy and hyperlipidemia were observed in both obese rats, and concomitantly, serum leptin and insulin were higher in fatty rats than in lean rats (Tables 1 and 2). It has been reported that leptin inhibits the production of triacylglycerol [45], whereas insulin promotes it [46,47]. In obese Zucker rats (fa/fa), approximately 2 to 4 times more triacylglycerol than in lean rats (+/+ or fa/+) was accumulated in the liver, followed by its secretion into circulation [48]. Since the leptin receptor did not function in the fatty rats, the turnover of triacylglycerol was more active in the synthesis than in the degradation by the action of insulin. Consequently, the accumulation of triacylglycerol was observed in the liver of obese rats. In the present study, this phenomenon occurred in ZDF-fatty rats as well as in Zucker-fatty rats, and this result reflected the amounts of insulin in sera well. In addition, the significantly lower total CoA levels in Zucker rats were clearly disadvantageous for β-oxidation, indicating that Zucker rats were in an in vivo environment that favored the accumulation of triacylglycerol (Table 3).

References

Shimabukuro, M.; Koyama, K.; Chen, G.; Wang, M.-Y.; Trieu, F.; Lee, Y.; Newgard, C.B.; Unger, R.H. Direct antidiabetic effect of leptin through triglyceride depletion of tissues. Proc. Natl. Acad. Sci. USA 1997, 94, 4637–4641. Foretz, M.; Guichard, C.; Ferré, P.; Foufelle, F. Sterol regulatory element binding protein-1c is a major mediator of insulin action on the hepatic expression of glucokinase and lipogenesis-related genes. Proc. Natl. Acad. Sci. USA 1999, 96, 12737–12742. Shimomura, I.; Bashmakov, Y.; Ikemoto, S.; Horton, J.D.; Brown, M.S.; Goldstein, J.L. Insulin selectively increases SREBP-1c mRNA in the livers of rats with streptozotocin-induced diabetes. Proc. Natl. Acad. Sci. USA 1999, 96, 13656–13661. Himeno, K.; Seike, M.; Fukuchi, S.; Masaki, T.; Kakuma, T.; Sakata, T.; Yoshimatsu, H. Heterozygosity for leptin receptor (fa) accelerates hepatic triglyceride accumulation without hyperphagia in Zucker rats. Obes. Res. Clin. Pract. 2009, 3, 1–52.

Reviewer 2-Comments to Author:

Comment 1: line 24-26, Sentence not clear, how would it help to elucidate features of obesity.

Response to #1: line 26-27, we added “in combination with result(s) of differential gene expression analysis and/or comparative genomics” to the end of this sentence as you suggested.

Comment 2: line 137, Which adipose tissues?

Response to #2: line 138-139, We have exchanged “the three adipose tissues” for “perirenal adipose tissues, brown adipose tissues, and epididymal adipose tissues”.

Comment 3: line 299, Which intermediates?

Response to #3: line 311, we modified this sentence: “in addition to their role as intermediates”.

Comment 4: line 312, in vivo italic?

Response to #4: We incorporated your suggestion, and the style of words have been changed to italic from plain as follows:

lines 54, 227, 231, 301, 306, and 309, “via”.

lines 174, 180, 245, 247, and 321, “i.e.”.

lines 262 and 324, “in vivo”.

Comment 5: line 312-314, how will it help to explain.

Response to #5: line 325-326, “together with differential gene expression analysis and/or comparative genomics” was added to the sentence.

Minor changes

line 25: We added “as” after “well”.

line 133: “drinking of” was exchanged with “drinking by”.

line 227: “the” was added after “via”.

Reviewer 2 Report

The manuscript by Chohnan et al. with the title ‘Coenzyme A and Its Thioester Pools in Obese Zucker 3 and Zucker Diabetic Fatty Rats’ is straight forward, well written and the experiments are well conducted.

Minor

24-26 Sentence not clear, how would it help to elucidate features of obesity?

137 which adipose tissues?

299 which intermediates?

312 in vivo  italic?

312-314 how will it help to explain…?

Author Response

(The authors gave the same response as above.)
